# Association of LTA and SOD Gene Polymorphisms with Cerebral White Matter Hyperintensities in Migraine Patients

**DOI:** 10.3390/ijms232213781

**Published:** 2022-11-09

**Authors:** Patrizia Ferroni, Raffaele Palmirotta, Gabriella Egeo, Cinzia Aurilia, Maria Giovanna Valente, Antonella Spila, Alberto Pierallini, Piero Barbanti, Fiorella Guadagni

**Affiliations:** 1Interinstitutional Multidisciplinary Biobank (BioBIM), IRCCS San Raffaele Roma, 00166 Rome, Italy; 2Department of Human Sciences and Quality of Life Promotion, San Raffaele Roma Open University, 00166 Rome, Italy; 3Interdisciplinary Department of Medicine, School of Medicine, University of Bari “Aldo Moro”, 70124 Bari, Italy; 4Headache and Pain Unit, IRCCS San Raffaele Roma, 00163 Rome, Italy; 5Department of Radiology, IRCCS San Raffaele Roma, 00163 Rome, Italy

**Keywords:** inflammation, oxidant stress, migraine, lymphotoxin α, superoxide dismutase, genetic variants, white matter hyperintensities

## Abstract

White matter hyperintensities (WMHs) in migraine could be related to inflammatory and antioxidant events. The aim of this study is to verify whether migraine patients with WMHs carry a genetic pro-inflammatory/pro-oxidative status. To test this hypothesis, we analyzed *lymphotoxin alpha* (*LTA*; rs2071590T and rs2844482G) and *superoxide dismutase 1* (*SOD1*; rs2234694C) and 2 (*SOD2*; rs4880T) gene polymorphisms (SNPs) in 370 consecutive patients affected by episodic (EM; n = 251) and chronic (CM; n = 119) migraine and in unrelated healthy controls (n = 100). Brain magnetic resonance was available in 183/370 patients. The results obtained show that genotypes and allele frequencies for all tested SNPs did not differ between patients and controls. No association was found between single SNPs or haplotypes and sex, migraine type, cardiovascular risk factors or disorders. Conversely, the *LTA* rs2071590T (OR = 2.2) and the *SOD1* rs2234694C (OR = 4.9) alleles were both associated with WMHs. A four-loci haplotype (*TGCT* haplotype: rs2071590T/rs2844482G/rs2234694C/rs4880T) was significantly more frequent in migraineurs with WMHs (7 of 38) compared to those without WMHs (4 of 134; OR = 8.7). We may, therefore, conclude by suggesting that that an imbalance between pro-inflammatory/pro-oxidative and antioxidant events in genetically predisposed individuals may influence the development of WMHs.

## 1. Introduction

Migraine is a primary form of headache, characterized by attacks of debilitating headache and symptoms of autonomic nervous system dysfunction. The most frequent types of migraine are episodic migraine without aura (MWoA) and migraine with aura (MWA); chronic migraine (CM, >15 days/month from at least from three months) derives from a worsening of pre-existing episodic migraine [1].

As migraine is a primary condition, brain imaging does not usually reveal any pathological alteration. In the last years, however, improvements to MRI-neuroimaging technology have permitted the disclosure of the presence of minute and focal white matter hyperintensities (WMHs) [2,3,4], represented by silent brain alterations in white cerebral substance that are located in supratentorial, subcortical and periventricular areas, especially in frontal lobes [5], although other studies have also supported the occurrence of WMHs in the posterior circulation [3]. These lesions have been linked to the course of migraine and an increased risk of WMHs has been observed in women or in patients with a longer disease duration, higher attack frequency or in the presence of comorbidities [6,7,8,9,10,11,12,13,14]. Accordingly, it is currently acknowledged that WMHs might be part of a gradual process, which could be responsible for the evolution of focal invisible microstructural changes into focal migraine-related visible WMHs [15].

Regarding their aetiology, the possibility of a vascular origin of WMHs was initially postulated on the basis of the increased risk of stroke and cardiovascular disease in migraine, particularly MWA [16,17,18], and was subsequently corroborated by the demonstration of a condition of peripheral vascular dysfunction, as suggested by the findings of an association between reduced carotid artery endothelial shear stress and a higher presence of WMHs in migraine compared to migraine-free controls [19]. However, the nature of these lesions, often detected as an occasional finding, is largely discussed and researchers are not in agreement about their aetiology and relation to the clinical characteristics of migraine and comorbidities. At present, the underlying mechanism(s) responsible for the development of WMHs remains unclear, although endothelial dysfunction [18,19], iron accumulation [3,20] and neuro-inflammation [21] have all been proposed as possible causes.

Despite all of the above, only few studies have investigated the association between biochemical signs of inflammation and the presence of WMHs in migraine. In particular, elevated levels of high-sensitive C-reactive protein (hs-CRP) have been reported in migraine patients but there was no correlation between WMHs and level of hs-CRP, suggesting that hs-CRP is not causally involved in the pathogenesis of WMHs in migraine patients [12]. 

Other authors have hypothesized that an increased oxidative stress and/or decreased antioxidant response may play a role in the pathophysiology of WMHs in migraine patients [22,23]. Indeed, a decreased antioxidant status in migraine patients with WMHs compared to those without WMHs and controls has been demonstrated by Aytaç et al. [22]. Accordingly, Erdélyi-Bótor et al. confirmed an impairment of the L-arginine/nitric oxide (NO) pathway demonstrating higher levels of known markers of endothelial dysfunction in migraine patients compared with controls, and suggested that elevated levels of the NO synthase inhibitor asymmetric dimethylarginine (ADMA) may impact the pathogenesis of migraine-related WMHs by influencing cerebrovascular autoregulation and vasomotor reactivity, while elevated L-arginine serum levels might reflect an increased demand for NO synthesis [23]. These changes would ultimately translate into a trigger for the development and progression of WMHs, which is in agreement with the current perspectives on the role of oxidative stress and/or decreased antioxidant defenses in migraine pathogenesis [24].

In this respect, Palmirotta et al. have postulated an impairment of oxidative pathway in migraine and suggested that *superoxide dismutase 2* (*SOD2)* is a disease-modifier gene influencing oxidative mechanisms in migraine [25]. In particular, the finding of an association between *SOD2* Ala16Val polymorphism and the occurrence of unilateral cranial autonomic symptoms (UAs) during the attack in MWA lead to the hypothesis that *SOD2* polymorphism may cause a defective control of the oxidative phenomena linked to cortical spreading depression (CSD), the neurophysiological hallmark of migraine aura, causing an overstimulation of trigeminal neurons and UAs triggering [25].

On the basis of these considerations, we hypothesized that patients with migraine could carry a particular genetic susceptibility responsible for a pro-inflammatory/pro-oxidative status that in some cases may lead to the development of WMHs. To test our hypothesis, we analyzed the presence of *lymphotoxin* α (*LTA*) and *SOD* genetic variants in patients with episodic and chronic migraine and correlated them with the presence of WMHs. The polymorphisms rs2234694 (c.239+34A>C) in intron 3 of *SOD1* and rs4880 (c.47T>C; p.Val16Ala) in exon 2 of *SOD2* genes were selected based on our previous observations [25]. The polymorphisms rs2844482 (c.-177-145C>T) and rs2071590 (c.-177-144A>G) in the promoter region of the *LTA* gene were analyzed, based on previous observation of their possible association with migraine phenotype [26,27]. In particular, the pedigree Genome-Wide Association Study (pGWAS) conducted in the genetic isolate Norfolk Island population identified five significant variants within the *TNF* gene cluster [27], one of which, located within the *LTA* gene (namely rs2844482), had been previously associated with migraine in a Korean population [26]. A number of other studies have also investigated genetic variants within the *LTA* gene and possible associations with migraine, with conflicting results [28,29]. Thus, in the present study only those SNPs with scientific evidence of an association with migraine characteristics were considered (rs2844482 and rs2071590) [26,27], but not those for which such evidence was lacking (e.g., rs2009658 or rs2229094) [28,29]. The results presented here add new scientific evidence of an association between specific *LTA* and *SOD1*/*SOD2* genotypes and the development of focal migraine-related visible WMHs, thus supporting the view that an imbalance between pro-inflammatory/pro-oxidative and antioxidant events in genetically predisposed individuals may influence the presence of WMHs, possibly conferring an increased risk of developing cerebrovascular accidents.

## 2. Results

Three hundred seventy migraine patients (MWoA 61, MWA 190, CM 119) and 100 unrelated healthy subjects were enrolled. Clinical and demographic characteristics of the enrolled patients are summarized in Table 1. As shown, patients with CM were most likely females of older age, but no differences were observed in the frequency of major cardiovascular risk factors or family history for cardiovascular disorders between episodic and chronic migraine.

*LTA*, *SOD1* and *SOD2* polymorphisms were analyzed in all 470 subjects, including 370 migraineurs and 100 unrelated controls. Genotypes and corresponding allele frequencies did not differ between all patients and controls (Table 2) and both groups were in Hardy–Weinberg equilibrium.

No association was found between single SNPs or haplotypes and sex, migraine type (MWA, MWoA and CM), or the presence of major cardiovascular risk factors (dyslipidemia, obesity, hypertension, smoking habit and type 2 diabetes), or overt cardiovascular disorders (11 cerebrovascular accidents and 2 myocardial infarctions) in patients with migraine. The haplotype analysis, considering a minimum frequency in either group of at least 2%, showed a presence in the overall population of 7 haplotypes with comparable prevalence between cases and controls, with the exception of haplotype *CAAC* (rs2071590, rs2844482, rs2234694 and rs4880), which exhibited a significant trend toward a protective effect in migraine (OR = 0.49 (95% C.I.: 0.25–0.98), *p* = 0.044) (Table 3). Linkage disequilibrium (LD) analysis, defined by the D’ coefficient as a measure of the non-random association of alleles at different loci, revealed a strong LD between the two *LTA* rs2071590 and rs2844482 polymorphisms (D’ = 0.96, r = 0.29), but not among *LTA* and *SOD1* or *SOD2* SNPs.

Cerebral MRIs were analyzed in a subgroup of 183 patients. To avoid confounding factors 11 patients with cerebrovascular disease were excluded. Post-hoc analysis showed that, beside a higher rate of CM patients—associated with a higher attack frequency (median 10 vs. 4 days/month, *p* < 0.0001)—the subgroup of patients who underwent MRI did not significantly differ from those who did not, both in terms of clinical/biomolecular variables or the presence of major cardiovascular risk factors. One hundred thirty-four patients (79 episodic migraine (64 MWoA, 15 MWA) and 55 with CM) had a normal cerebral MRI. The presence of WMHs was detected in 38 patients (13 episodic migraine (9 MWoA, 4 MWA) and 25 with CM). The genotypes and the allelic frequencies did not significantly deviate from the Hardy–Weinberg equilibrium between the two groups of patients that underwent MRI. 

SNP analysis of observed allele and genotype frequencies demonstrated that the *LTA* rs2071590 T allele (OR = 2.2; *p* = 0.034) and the *SOD1* rs2234694 C allele (OR = 4.9; *p* = 0.003) were both associated with MRI findings of WMHs in inheritance models of migraine patients (Table 4). Moreover, as shown in Table 3, the frequency of the rs2071590, rs2844482, rs2234694 and rs4880 *TGCT* haplotype was significantly higher in migraineurs with WMHs at MRI compared to those without (OR = 8.7, *p* = 0.009) (Table 3). In particular, the *TGCT* haplotype was present in 7 (18.4%) of 38 patients with signs of WMHs at MRI compared to 4 (3.0%) of 134 patients without (Fisher test: *p* = 0.003). The prevalence of the other observed haplotypes was comparable between patients with or without WMHs.

Multivariate logistic regression analysis of independent determinants of WMHs in migraine patients confirmed that a *TGCT* haplotype carrier status (together with chronicization and advanced age) was independently associated with an increased risk of WMHs at MRI (Table 5).

## 3. Discussion

Despite the strong recommendation to avoid neuroimaging in migraine sufferers as part of a normal neurologic examination [30], a large proportion of patients still undergo inappropriate CT or MRI brain imaging without any reliable clinical reason [31]. The accidental identification of WMHs is often misleading, resulting in disease misconception. Thus, a better comprehension of the pathophysiology of WMHs is needed to avoid unnecessary diagnostic investigations, ensuring proper and early migraine management [32]. 

Migraine is a chronic cortical brain disorder with paroxysmal manifestations. The mismatch between hyperexcitability and hypometabolism represents the pathophysiological substrate of the migraine brain [33]. In this scenario, periodical fluctuations of cortical excitability, plasticity, and metabolism due to the impact of external or internal triggers ignite the trigemino-vascular system, leading to development of migraine headache. The release of neuropeptides as substance *p*, and calcitonin gene-related peptide (CGRP) from sensitized first-order nociceptive neurons induces a neurogenic inflammatory response in the meninges that involves vascular cells, immune cells and neurons [34,35]. In patients affected by MWA, the neuroinflammatory response spreads to the parameningeal tissues (meninges and bone marrow) facing the cortical areas generating CSD, as reported in a simultaneous 11C-PBR28PET/magnetic resonance imaging (MRI) study [36]. 

In experimental migraine models, the brain activates genes coding for proteins protecting against oxidative and inflammatory events during the CSD (e.g., major prion protein, glutathione-S-transferase-5, and apolipoprotein E), hinting that under normal conditions the brain tries to counteract potentially harmful consequences of a benign headache disorder [37]. It is noteworthy that ineffective acute treatment increases the risk of developing chronic migraine, supporting the view that prolonged brain exposure to the inflammatory and oxidative events induced by the activation of the trigeminal sensory pathway would contribute to migraine chronicization [38].

All the above data suggest that an imbalance between pro-inflammatory/pro-oxidative and antioxidant events in genetically predisposed individuals may influence the presence of WMHs. This hypothesis is corroborated by the results obtained in the present study showing an association between specific *LTA* and *SOD1*/*SOD2* genotypes and the development of focal migraine-related visible WMHs.

*LTA* (OMIM *153440), also known as *TNFB*, encodes for a cytokine capable of modulating numerous inflammatory, immunological and antiviral responses. It has been postulated that inflammatory processes modulated by *LTA* may contribute to the propagation of neuronal hyperexcitability by acting as an initiating and maintaining factor during migraine attacks [39]. Indeed, several studies have shown an association between *LTA* gene polymorphisms and susceptibility to migraine, but results have often been controversial [26,28,29,40,41,42,43]. Actually, previous studies performed on 76 and 685 Australian Caucasian migraine patients showed no association with *LTA* rs2844482 and rs2071590 respectively [28,29], which is in agreement with the findings obtained in the present study that show no differences between *LTA* genotypes and corresponding allele frequencies between patients and controls. Furthermore, we could not demonstrate any association between *LTA* SNPs and sex, migraine type, the presence of major cardiovascular risk factors, or overt cardiovascular disorders. On the other hand, the haplotype analysis conducted in the present study demonstrated that the rs2071590 and rs2844482 GT (CA) haplotypes were significantly associated with a protective effect against migraine and both were in strong LD. These results are in agreement with a study conducted on 439 Korean migraine patients, genotyped for some polymorphisms of the *LTA* gene, showing that the presence of the C allele (G) rs2844482 correlated significantly with increased risk of all types of migraine, while rs2071590 A allele (T) was more represented only in patients with MWA [26].

However, the novel aspect of our study was the combined analysis of *LTA* and *SOD1*/*SOD2* gene polymorphisms. *SOD1* (OMIM *147450) and *SOD2* (OMIM *147460) genes encode for a cytoplasmic protein that exerts most of the cellular superoxide dismutase activity and a mitochondrial protein deputed to limit superoxide accumulation from oxidative metabolism respectively. In a previous study on 97 Turkish children and adolescents, Saygi and colleagues observed the presence of a significant correlation between *SOD2* rs4880 TT genotype and disease [44], a finding that could not be fully reproduced in a subsequent study by Palmirotta et al., who suggested that *SOD2* may act as a disease-modifier gene influencing oxidative mechanisms in migraine, based on the finding of an association between *SOD2* Ala16Val polymorphism and the occurrence of UAs during the attack in MWA [25]. Here, we confirm our previous findings and, for the first time to our knowledge, demonstrate that the *LTA* rs2071590 T allele (OR = 2.2) and the *SOD1* rs2234694 C allele (OR = 4.9) were both associated with MRI findings of WMHs in migraine patients. Of interest, haplotype frequency estimation in the overall population of migraine patients showed the presence of a four-loci (rs2071590/rs2844482/rs2234694/rs4880) haplotype (*TGCT* haplotype), the frequency of which was higher in migraine patients with WMHs at MRI (18.4%) compared with those without (3.0%) and was significantly associated with WMHs both in haplotype association studies (OR = 8.7) and in a multivariate logistic regression analysis model (OR = 13.9).

There are, of course, some limitations to acknowledge. First, the sample size was relatively small, and so might have led to an overestimation of the magnitude of observed associations. Secondly the study was of a retrospective nature and recruitment was carried out in a single institution, which might have hampered its external validity. Lastly, imaging was available only in a limited subgroup of migraine patients and no MRI was performed in the control group. On the other hand, the main strength resides in the fact that the study was designed on a dataset derived from a biobank/database project in which patients’ thorough demographic and clinical characterization was performed by specifically trained neurologists using detailed face-to-face interviews. Therefore, the population enrolled was highly homogenous and well characterized. With that in mind, the results reported here suggest that an imbalance between pro-inflammatory/pro-oxidative and antioxidant events in genetically predisposed individuals may influence the course of migraine.

To the best of our knowledge, this is the first evidence of an association between *LTA* and *SOD1*/*SOD2* genotypes and the development of focal migraine-related visible WMHs. At present, we may theorize that genetic predisposition may elicit an adaptive response (with disruption of oxidative homeostasis and/or cytokine upregulation) mostly in individuals who carry the *TGCT* haplotype. The resulting imbalance between pro-inflammatory/pro-oxidative and antioxidant events would ultimately translate into a trigger for the evolution of focal invisible microstructural changes into focal migraine-related visible WMHs. A causal association of increasing WMHs burden with stroke has been recently demonstrated both in a Mendelian randomization study [45] and a large meta-analysis of more than 16,000 participants [46]. As migraine is associated with a two-fold increase in the risk of developing ischemic stroke, it is, therefore, conceivable to hypothesize that the presence of WMHs might represent a possible neuroimaging biomarker to identify migraine patients at increased risk of developing cerebrovascular accidents. However, this hypothesis is merely speculative and requires detailed experimental evaluation before its ultimate significance can be determined. Further multicenter prospective and validation cohort studies involving larger numbers of patients are needed to fully establish their role in migraine and associated comorbidities.

## 4. Materials and Methods

### 4.1. Patients and Sample Collection

Starting from January 2008, the Headache and Pain Unit and the InterInstitutional Multidisciplinary Biobank (BioBIM) of the IRCCS San Raffaele Rome, Italy, have been jointly involved in the recruitment of outpatients affected by headache, who are then prospectively followed under the appropriate institutional ethics approval and in accordance with the principles embodied in the Declaration of Helsinki [47]. All patients routinely undergo general and neurological examinations, are interviewed with a face-to-face semi-structured questionnaire detailing in depth the demographic and clinical characteristics of migraine and provide written informed consent to donate a blood sample to be used in the analysis of possible determinants of migraine. Blood samples are collected during the outpatient visit in fasting conditions and processed using standard operating procedures, ICT tools and dedicated software to track the entire sample life, including elapsed time between blood withdrawal and storage [47]. Plasma and serum samples are aliquoted, coded, and stored at −80 °C for subsequent batch analyses. Storage conditions are carefully monitored, and all aliquots are limited to one freeze-thaw cycle to ensure the best sample quality.

For the present study, 370 unrelated Caucasian patients affected by MWA (n = 61), MWoA (n = 190) and CM (n = 119). Clinical and demographic characteristics of the studied population are reported in Table 1.

Cerebral magnetic resonance imaging (1.5 T) was available from our department of radiology in a subgroup of 183 patients. All MRI were evaluated by a neuro-radiologist, specialized in headache neuroimaging, blinded to migraine diagnosis. Eleven patients with cerebrovascular disease were excluded to avoid any bias related to comorbidity confounding factors. Thus, final association studies between biomolecular findings and imaging data on WMHs were performed on 172 patients (Figure 1).

As a control group, 100 unrelated non-migraine individuals (40% male, 60% female, aged 38.4 ± 12.2 years) from the same geographical area of the patients were recruited from apparently healthy individuals enrolled in the BioBIM in order to evaluate and compare genotype/haplotype frequencies in their ethnicities.

### 4.2. Molecular Analysis of Lymphotoxin Alpha (LTA) and Superoxide Dismutase (SOD) Gene Polymorphisms

All samples to be tested in subsequent molecular analyses were processed and appropriately cryopreserved following the BioBIM standard operating procedures [48].

The polymorphisms rs2844482 (c.-177-145C>T) and rs2071590 (c.-177-144A>G) in the promoter region of *LTA*, rs2234694 (c.239+34A>C) in intron 3 of *SOD1* and rs4880 (c.47T>C; p.Val16Ala) in exon 2 of *SOD2* genes were analyzed in all 370 patients and 100 controls. Genomic DNA was isolated from the peripheral blood using the DNeasy^®^ blood and tissue Kit (QIAGEN Inc., Chatsworth, CA, USA) according to the manufacturer’s protocol and quantified with a Qubit^®^3.0 fluorometer (Life Technologies™, Carlsbad, CA, USA).

Standard PCR was performed using the following primers: F5′-ACTGCTGTTTCAGTCAAAGGC-3′ (LTA)R5′-ATGATTGCTCTTCAGGGAACC-3′ (LTA)F5′-TATCCAGAAAACACGGTGGGCC-3 (SOD1)R5′-TCCTGTATTAGTTCCCCTTTGGCAC-3 (SOD1)F5′-TCTCGTCTTCAGCACCAGCAGG-3′ (SOD2)R5′-TGGTACTTCTCCTCGGTGACG-3′ (SOD2)
in a GeneAmp PCR System 9700 (Applied Biosystems, Foster City, CA, USA) using HotStarTaq Master Mix (HotStarTaq Master Mix Kit, QIAGEN Inc., Chatsworth, CA, USA). Specific sequencing reactions with the same pairs of primers were performed using a Big Dye Terminator v3.1 Cycle Sequencing kit (Applied Biosystems, CA, USA) on a 3500 Series Genetic Analyzer (Thermo Fisher Scientific Inc., Foster City, CA USA). In order to exclude pre-analytical and analytical errors PCR reactions and sequencing analyses were carried out on two different DNA extractions and sequencing analyses were performed on both strands. 

For primer design and to compare the sequence results we referred to the Ensembl project (https://www.ensembl.org (accessed on 20 April 2022)) sequences *LTA* (ENST00000418386.3), *SOD1* (ENST00000270142.11) and *SOD2* (ENST00000538183.7) [49]. 

### 4.3. Statistical Analysis

Data are presented as numbers, percentages, mean + SD, or median and interquartile ranges (IQR). Allelic frequencies were estimated by gene counting and genotypes were scored. Frequencies of each SNP genotype were compared with those expected for a population in Hardy–Weinberg equilibrium. The significance of the differences of observed alleles and genotypes between groups, as well as haplotype frequencies and associations, were tested using a free web-based application for association studies in multiple inheritance models [50]. Akaike information criterion (AIC) was used to determine the best-fitting inheritance model for analyzed SNPs, with the model with the lowest AIC reflecting the best balance of goodness-of-fit and parsimony [51].

Differences between percentages were assessed by chi-square test. Student’s unpaired t test and ANOVA test were used for normally distributed variables. Appropriate non-parametric tests (Mann–Whitney U test and Kruskal–Wallis test) were employed for all the other variables. Logistic regression analysis was used to evaluate the association between gene polymorphisms, clinical covariates and the presence of WMHs.

Population size was based on sample availability rather than sample size calculations. Only two-tailed probabilities were used for testing statistical significance and *p* values lower than 0.05 were regarded as statistically significant. Calculations were made using a computer software package (Statistica 8.0, StatSoft Inc., OK, USA) or free web-based applications (http://statpages.org/ (accessed on 20 April 2022)).

## 5. Conclusions

The results here presented add new scientific evidence of an association between specific *LTA* and *SOD1*/*SOD2* genotypes and the development of focal migraine-related visible WMHs, thus supporting the view that an imbalance between pro-inflammatory/pro-oxidative and antioxidant events in genetically predisposed individuals may influence the presence of WMHs, possibly conferring an increased risk of developing cerebrovascular accidents.

## Figures and Tables

**Figure 1 ijms-23-13781-f001:**
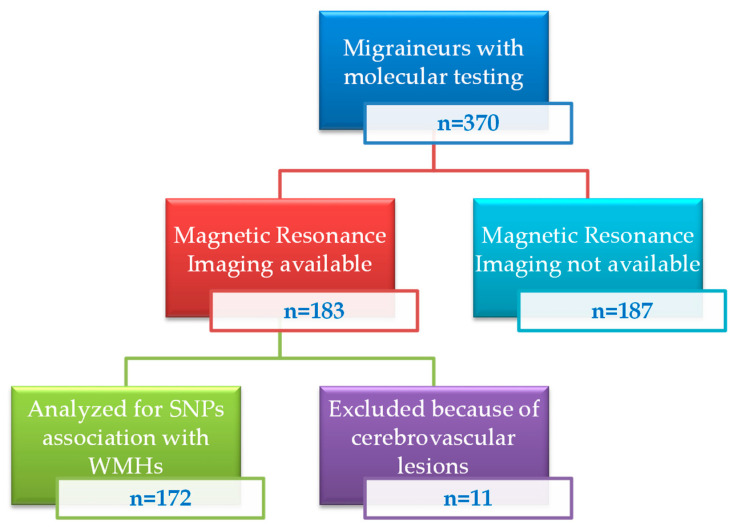
Flow chart of patients’ recruitment.

**Table 1 ijms-23-13781-t001:** Patients’ clinical and demographic characteristics.

	MWA	MWoA	CM	*p* Value *
N. enrolled	61	190	119	
Males/Females	15/46	39/151	11/108	<0.001
Age	38.7 ± 12.3	40.2 ± 11.5	47.3 ± 13.3	<0.001
Migraine frequency (days/month)	3.0 ± 3.6	5.2 ± 3.8	24.1 ± 8.2	<0.001
Dyslipidemia	7 (11.5%)	15 (7.9%)	18 (15.1%)	0.113
Obesity (BMI > 30)	3 (4.9%)	7 (3.7%)	4 (3.4%)	1.000
Hypertension	13 (21.3%)	10 (5.3%)	14 (11.8%)	0.529
Smoking habit	17 (27.9%)	50 (26.3%)	28 (23.5%)	0.574
Diabetes	1 (1.6%)	3 (1.6%)	2 (1.7%)	1.000
Family history ^1^	16 (26.2%)	32 (16.8%)	28 (23.5%)	0.360

^1^ Family history for cardiovascular disorders. MWoA, migraine without aura; MWA, migraine with aura; CM, chronic migraine; BMI, body mass index. * *p* value is reported for differences between episodic and chronic migraine. Two-tailed Fisher exact test is given for categorical variables. Mann–Whitney test was used for continuous variables.

**Table 2 ijms-23-13781-t002:** *Lymphotoxin alpha* (*LTA*) and *superoxide dismutase 1* (*SOD1*) and 2 (*SOD2*) genetic polymorphisms and corresponding allele frequencies in 370 migraine patients and 100 control subjects.

						Genotype Frequencies							
		Allele Frequencies				Control Subjects				Migraine Patients			
Rs	**SNP**	Controls		Patients		WT	HET	HOM	*p **	WT	HET	HOM	*p **
* **LTA** *													
**rs2071590**	**C→T**	0.66	0.34	0.62	0.38	0.44	0.44	0.12	*0.83*	0.37	0.50	0.12	0.18
**rs2844482**	**A→G**	0.17	0.83	0.13	0.87	0.04	0.26	0.70	0.47	0.01	0.24	0.75	0.49
* **SOD1** *													
**rs2234694**	**A→C**	0.94	0.06	0.95	0.05	0.90	0.09	0.01	0.43	0.89	0.10	0.01	1.00
* **SOD2** *													
**rs4880**	**C→T**	0.44	0.56	0.45	0.55	0.23	0.43	0.34	0.22	0.22	0.46	0.32	0.21

* *p* value for Hardy–Weinberg equilibrium. SNP, single nucleotide polymorphism; WT, wild type; HET, heterozygous; HOM, homozygous.

**Table 3 ijms-23-13781-t003:** *Lymphotoxin alpha* (*LTA*) and *superoxide dismutase* 1 (*SOD1*) and 2 (*SOD2*) haplotype analysis and corresponding frequencies in 370 migraine patients and 100 control subjects.

Haplotypes						Migraine Patients			
*LTA rs2071590*	*LTA* rs2844482	*SOD1* rs2234694	*SOD2* rs4880	Controls (n = 100)	Migraine Patients (n = 370)	Without WMHs (n = 134)	With WMHs (n = 38)	OR (95% C.I.)	*p* *
C	G	A	T	0.28	0.28	0.28	0.20	1.00	NA
C	G	A	C	0.16	0.20	0.21	0.21	0.34 (0.43–4.22)	0.62
T	G	A	T	0.19	0.19	0.19	0.12	1.75 (0.58–5.31)	0.32
T	G	A	C	0.14	0.17	0.17	0.21	1.04 (0.38–2.79)	0.94
C	A	A	C	0.11	0.05 **	0.07	0.08	1.68 (0.51–5.57)	0.40
C	A	A	T	0.06	0.06	0.04	0.05	1.95 (0.31–12.2)	0.47
T	G	C	T	0.02	0.04	0.01	0.09	8.68 (1.76–42.8)	0.009

* *p* value for haplotype association with white matter hyperintensities (WMHs). ** Migraine vs. controls: OR = 0.49 (95% C.I.: 0.25–0.98), *p* = 0.044.

**Table 4 ijms-23-13781-t004:** *Lymphotoxin alpha* (*LTA*) and *superoxide dismutase* 1 (*SOD1*) and 2 (*SOD2*) genetic polymorphisms and corresponding genotype frequencies in 172 migraine patients with (n = 38) or without (n = 134) white matter hyperintensities (WMHs).

						Genotype Frequencies *								
		Allele Frequencies				Without WMHs			With WMHs					
Rs	SNP	Without WMHs		With WMHs		WT	HET	HOM	WT	HET	HOM	OR (C.I.)	AIC	*p*
*LTA*														
rs2071590	C→T	0.61	0.39	0.55	0.45	0.37	0.49	0.14	0.21	0.68	0.11	2.23 (1.04–4.79)	181	0.034 ^†^
rs2844482	A→G	0.12	0.88	0.14	0.86	0.02	0.20	0.78	0.00	0.29	0.71	1.69 (0.74–3.85)	184	0.22
*SOD1*														
rs2234694	A→C	0.97	0.03	0.87	0.13	0.94	0.06	0.00	0.76	0.21	0.03	4.89 (1.74–13.8)	177	0.003 ^‡^
*SOD2*														
rs4880	C→T	0.51	0.49	0.45	0.55	0.27	0.43	0.30	0.21	0.47	0.32	0.92 (0.42–2.01)	185	0.84

* Eleven patients with cerebrovascular lesions were excluded from the analysis. ^†^ *p* value for SNP association with WMHs in an over-dominant model. ^‡^ *p* value for SNP association with WMHs in a dominant model. SNP, single nucleotide polymorphism; WT, wild type; HET, heterozygous; HOM, homozygous; AIC: Akaike information criterion.

**Table 5 ijms-23-13781-t005:** Logistic regression analysis of independent determinant of white matter hyperintensities (WMHs) in migraine patients.

	Enter Model			Stepwise		
Variable	OR	95% CI	*p*	OR	95% CI	*p*
Migraine type *	5.01	1.07–23.5	0.041	2.48	1.06–5.82	0.037
*TGCT* haplotype	14.4	3.08–67.8	0.001	13.9	3.23–59.7	0.001
Age	1.04	1.01–1.08	0.021	1.04	1.01–1.07	0.017
Smoking habit	2.77	1.08–7.11	0.035	2.44	0.99–6.02	0.052
Aura	2.29	0.52–10.0	0.270			
Diabetes	0.00	0.00–0.00	0.994			
Dyslipidemia	0.86	0.25–2.93	0.809			
Hypertension	0.88	0.20–3.77	0.859			
Body mass index	1.10	0.96–1.27	0.170			
Frequency	0.98	0.92–1.05	0.547			
Sex	0.58	0.16–2.19	0.424			

* Coded as episodic/chronic, 0/1.

## Data Availability

The data presented in this study are available on request from the corresponding author.

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
