# Peer review of "Association of LTA and SOD Gene Polymorphisms with Cerebral White Matter Hyperintensities in Migraine Patients"

_ijms, 2022, doi:10.3390/ijms232213781_

Round 1

Reviewer 1 Report

Make a rigorous satistical analysis

1. Add questionnaire at the end of the paper

2. Calculate the cronbach alpha of the constructs

3. Apply EFA on the questionnaires and see which items are not contributing and mentioned them in the paper, that why are not they contributing and reasons behind it.

Improve the introduction by adding a critical literature review.

Author Response

We apologize, but the meaning of this comment is not clear to us. It looks like there is a misunderstanding on the nature of the study, which is not dealing with a survey of self-reported questionnaires. The questionnaire mentioned in the patients section is a shared questionnaire, filled out by the specifically-trained board certified neurologists during face-to-face interviews. This semistructured questionnaire - routinely used by our research group for both clinical and scientific aims - covers socio-demographic features, lifestyle, migraine characteristics and treatments, concomitant therapies and comorbidities. It has been used in all our scientific papers and consists of approximately 100 items that are used to populate the migraine registry, and whose description is beyond the object of the study. Said that, a rigorous statistical analysis based on Cronbach’s Alpha calculation of the constructs and EFA application on the questionnaire survey is not applicable to the study here presented, in which inheritance models, haplotype analysis and logistic regression were used to test associations between SNPs and radiological findings.

Reviewer 2 Report

The authors presented a study to investigate the association between LTA and SOD1/SOD2 genotypes and the development of focal migraine-related visible WMHs. They concluded that an imbalance between pro-inflammatory/pro-oxidative and antioxidant events in genetically predisposed individuals may influence the development of WMHs. This is, in my opinion, an interesting paper on quite a novel argument. However, there are some minor points to be further improved as well.

1. P4, Line 132-134, the results showed “No association was found between single SNPs or haplotypes and sex, migraine type (MwA, MWoA and CM), or the presence of major cardiovascular risk factors (dyslipidemia, obesity, hypertension, smoking habit and type 2 diabetes), or overt cardiovascular disorders (11 cerebrovascular accidents and 2 myocardial infarctions)”. --- Table 1 depicts the demographic characteristics of patients in three diagnostic groups. where is the p-value? 

2. The abbreviations "WT', "HET, and "HOM" in Table 2 should be explained in the remarks at the bottom of the table.

Author Response

We thank the Reviewer for kind appreciation of our study and for valuable suggestions to improve the manuscript.

  1. P4, Line 132-134, the results showed “No association was found between single SNPs or haplotypes and sex, migraine type (MwA, MWoA and CM), or the presence of major cardiovascular risk factors (dyslipidemia, obesity, hypertension, smoking habit and type 2 diabetes), or overt cardiovascular disorders (11 cerebrovascular accidents and 2 myocardial infarctions)”. --- Table 1 depicts the demographic characteristics of patients in three diagnostic groups. where is the p-value?

We apologize for the misunderstanding, but the results of the univariate analysis were not shown, as no association was found. This has been clarified in the text (page 4, line 149). Concerning Table 1, a column has been added reporting the significance level between episodic and chronic migraine, legend has been implemented accordingly and results have been discussed in the text (page 3, lines 122-125).

  1. The abbreviations "WT', "HET, and "HOM" in Table 2 should be explained in the remarks at the bottom of the table.

Text has been emended according to the reviewer suggestion.

Reviewer 3 Report

The subject of WML in migraine has been a troubling matter since MRI has been more accessible. This is a very nice study trying to link polymorphisms to the appearance and increase in number (longitudinal studies) of WMLs.

There are some points that the authors need to consider:

1.     Page 1 Line 42: doesn't: Change to does not.

2.     Page 2 Line 47:  especially in frontal lobes: This fact is under dispute. There are studies that support lesions in the posterior circulation. https://doi.org/10.1111/j.1468-2982.2009.01904.x  Please rephrase.

3.     Page 2 Lines 98-100: The polymorphisms rs2234694 (c.239+34A>C) in intron 3 of SOD1 and rs4880 (c.47T>C; p.Val16Ala ) in exon 2 of SOD2 genes were selected based on our previous observations.: Reference needed about the observations because all the hypothesis testing is based on this.

4.     Page 2-3 Lines 100-102: The polymorphisms rs2844482 (c.-177-145C>T) and rs2071590 (c.-177-144A>G) in the promoter region of the LTA gene were analyzed, based on previous observation of their possible association with migraine phenotype.: Reference needed about the observations because all the hypothesis testing is based on this. Why didn't you use rs2009658 and rs2229094 also as Oikari et al did? DOI: 10.1016/j.gene.2012.09.116

5.     TABLES: Write abbreviations below the tables: eg WT wild type, HET, HOM , SNP etc

6.     Page 4 Lines 132-135: Was this association made for the migraine (case) group? If yes please clarify.

7.     Limitations: Please add to the limitations the fact that in this case control study no MRI and analysis was performed to the control group.

8.     Would you consider mentioning other gene polymorphisms (of white matter lesions) to the discussion, such as htra1?

Otherwise a nicely executed study.

Author Response

We thank the Reviewer for kind appreciation of our study and for valuable suggestions to improve the manuscript.

There are some points that the authors need to consider:

  1. Page 1 Line 42: doesn't: Change to does not.

Text has been emended according to the reviewer suggestion.

  1. Page 2 Line 47: especially in frontal lobes: This fact is under dispute. There are studies that support lesions in the posterior circulation. https://doi.org/10.1111/j.1468-2982.2009.01904.x  Please rephrase.

Text has been rephrased according to the reviewer suggestion and citation has been quoted (page 2, lines 47-48).

  1. Page 2 Lines 98-100: The polymorphisms rs2234694 (c.239+34A>C) in intron 3 of SOD1 and rs4880 (c.47T>C; p.Val16Ala ) in exon 2 of SOD2 genes were selected based on our previous observations.: Reference needed about the observations because all the hypothesis testing is based on this.

Text has been emended according to the reviewer suggestion and reference has been quoted at the end of the sentence (page 3, line 101).

  1. Page 2-3 Lines 100-102: The polymorphisms rs2844482 (c.-177-145C>T) and rs2071590 (c.-177-144A>G) in the promoter region of the LTA gene were analyzed, based on previous observation of their possible association with migraine phenotype.: Reference needed about the observations because all the hypothesis testing is based on this. Why didn't you use rs2009658 and rs2229094 also as Oikari et al did? DOI: 10.1016/j.gene.2012.09.116

Text has been emended according to the reviewer suggestion and references have been quoted at the end of the sentence (page 3, lines 109-112). Specifically, we did not test the rs2009658 and rs2229094 SNPs because Oikari et al. (reference 29) demonstrated no association with migraine in the Australian Caucasian case-control population examined. This lack of association was confirmed by Stuart et al. (reference 30), who showed no association with migraine susceptibility in a large Caucasian population. Thus, we limited our study to rs2844482 (shown to significantly associate with migraine) and rs2071590 (shown to significantly associate with migraine susceptibility) as reported by Lee et al. (reference 27).

  1. TABLES: Write abbreviations below the tables: eg WT wild type, HET, HOM , SNP etc

Tables have been emended according to the reviewer suggestion.

  1. Page 4 Lines 132-135: Was this association made for the migraine (case) group? If yes please clarify.

Text has been emended according to the reviewer suggestion (page 4, line 149).

  1. Limitations: Please add to the limitations the fact that in this case control study no MRI and analysis was performed to the control group.

Text has been emended according to the reviewer suggestion (page 7, lines 274-276)

  1. Would you consider mentioning other gene polymorphisms (of white matter lesions) to the discussion, such as htra1?

This would be a very interesting topic to be addressed. However, in our opinion a discussion of other gene polymorphisms (of white matter lesions) would very highly speculative and not substantiated by any experimental finding. Concerning specifically the HTRA1 gene, beside the fact that we do not have any data, and despite its suggested association with WMHs in other clinical settings, to the best of our knowledge there is only one report in migraine patients, showing no association. In light of these considerations, we do not believe that this is the most suitable context for discussing this topic. We hope the reviewer will agree with our considerations. If not, we are ready to further revise the document accordingly.